# Invited perspective: "Natural hazard management, professional development and gender equity: let's get down to business."

Valeria Cigala[1][†], Giulia Roder[2][†], Heidi Kreibich[3]

[1]Department of Earth and Environmental Sciences, Ludwig-Maximilians-Universität München, Munich, 80799, Germany

[2]Department of Economics and Statistics, University of Udine, Udine, 33100, Italy

[3]GeoForschungsZentrum Potsdam (GFZ), Section Hydrology, Telegrafenberg, D-14473 Potsdam, Germany

[†] Contributed equally

*Correspondence to*: Valeria Cigala (valeria.cigala@min.uni-muenchen.de) and Giulia Roder (giulia.roder@uniud.it)

OrcIDs

Valeria Cigala 0000-0003-2410-136X

Giulia Roder 0000-0002-0644-3625

Heidi Kreibich 0000-0001-6274-3625

**1 Take stock of the situation**

Women constitute a minority in the geoscience professional environment (around 30%, e.g., UNESCO, 2015; Gonzales, 2019; Handley et al., 2020), and as a consequence, they are underrepresented in disaster risk reduction (DRR) planning. After examining the Sendai framework documents and data outputs, Zaidi and Fordham (2021) pointed out that the Sendai Framework for Disaster Risk Reduction 2015–2030 (SFDRR) has failed to promote women and girls' inclusion in disaster policy effectively. In addition, it represents a missed opportunity to tackle gender-based issues in DRR (even beyond the female-male dichotomy). Nevertheless, practical actions have been promoted and applied in several contexts with promising results, but often they only remain lessons learned in localised environments (Zaidi and Fordham, 2021). Instead, the global gender gap index, which includes political empowerment, economic participation and opportunity, educational attainment, health, and survival, reveals that the average distance completed to parity is only 68% in 2019. Although the gap closing rate has constantly improved, it will take about 135.6 years to close it completely (WEF, 2021). These numbers do not yet account for 2020-2021 data, where the global pandemic has more strongly impacted women, their career, their opportunities, and their health in comparison with men (e.g., Alon et al., 2020; Chandler et al., 2021; Yildirim and Eslen-Ziya, 2021).

Gender recognition and representation do not affect the sole career sphere or the policy and DRR agenda. They even impact our vision about gender and gender equity in the actions, behaviours, and intentions before, during and after

natural hazards. Based on our literature search, we recognise that for most disaster-related papers, gender was merely
used as a dichotomous variable (usually together with a set of other socio-demographic variables) to test assessments
and model results, which are the core of the papers. When gender results in a significant variable, it is rarely
contextualised with the vulnerability of women and men in the socio-cultural and political environment of the study
site (exceptions are e.g., Finucane et al., 2000; Cvetkovic et al., 2018; Mondino et al., 2021). Instead, stereotypical
biological sex motivations are more often considered (e.g., women are more vulnerable due to housekeeping and child-
bearing responsibilities (Paradise, 2005; De Silva and Jayathilaka, 2014)). Gender as a social structure has a complex
interaction both at the individual and communal levels (Risman, 2018), able to influence the capacity of communities
to withstand the negative occurrence of natural hazards actively. In our opinion, if we fail to understand that, we fail
in risk reduction strategies and effective planning. To this point, we recognise that gender is poorly investigated in
DRR papers. It is much more considered in social sciences articles, oriented to history, societies, and social behaviours
in general. Moreover, gender diversity is scarce in the professional sphere of natural hazards, with consequences for
managing vulnerabilities and career opportunities in academic research.
Thus, despite the global gender gap index decreasing over the years, challenges to gender equity (e.g. reaching equal
political power, economic participation, educational attainment) are still strongly perceived. Therefore, practical
actions, solutions, and strategies to close the gender gap must continue to be tested and researched, the actions' efficacy
assessed, and their effects adequately monitored. In this 'invited perspective', we put individuals identifying
themselves with genders that are a minority in the field of natural hazards, i.e. female and non-binary genders, at the
centre of the discussion. We aim to concretely contribute to understanding the standpoint of these minorities who are
often underrepresented, unheard and poorly considered professionally in DDR policy and practice. Thus, this
perspective qualitatively explores a collection of 121 opinions of individuals identifying themselves as female and
one opinion of an individual identifying themselves as non-binary working in the broad field of natural hazards (in
academia, in the industry, as practitioners or policymakers). The respondents are disproportionate towards the female
gender; as a result, most of the issues and solutions proposed and discussed in the present paper revolve around the
female gender.
The questionnaire was short and explorative, examining opinions on the challenges (Q1) related to natural hazards in
general and those concerning (Q2) natural hazards and gender equity, plus (Q3) on the most urgent solutions to
withstand gender inequities. The last question (Q4) asked for the respondent's gender-related challenges experienced
during their career (or studies). Questions have been purposely developed following a general-to-local scale,
narrowing down their general perspectives in natural hazards research and concluding with one's own experience. We
have chosen open questions to let the professionals personally provide the most critical priority for action, related
challenges, and solutions. We have categorised all the answers through qualitative text analysis. Each response to the
four questions has been analysed independently by the three authors. A final discussion allowed to assign all responses
to definitive categories to the key concepts expressed. All categories are shown in Figure 1. The survey included socio-
demographic variables (profession, educational level, and country of residence) characterising the respondents. The
data collection used a random approach, where only interested participants offered their time participating in the
survey; we found a heterogeneous (and disproportionate) representation of those demographic categories. The survey
was conducted in April 2021 online on EUSurvey, a service created and managed by the European Commission. The
survey was fully anonymised, and no user-related data were saved. No respondent's sensitive information (e.g., name,
surname or age) was asked. The survey, i.e. link to the questionnaire with a short explanatory and motivational text,
was advertised via email to the EGU NHESS author list and to a list of female professionals that the authors had
collected in their networks. Moreover, the survey was advertised on social media, particularly on Twitter, LinkedIn,
and Facebook, through the personal accounts of the first two authors.
Among 122 people who filled the questionnaire, 121 recognised themselves as female and one as non-binary. Since
also non-binary people are underrepresented, we decided to include their answer in the analysis. Table 1 summarises
the demographics of the respondents. Individuals recognising themselves as male were excluded from the survey via
a first barrier question about the gender. The sample is dominated by female, European scientists working on hydro-
meteorological hazards or multi-hazards.
*Table 1. Summary of the respondents' demographics expressed in percentage.*

| Identified gender | Respondents [%] |
|---|---|
| Female | 99.2 |
| Non-binary | 0.8 |
| **Natural Hazard field** | |
| Hydro-meteo | 39.3 |
| All or multiple | 26.2 |
| Landslides | 13.9 |
| Earthquakes | 9.0 |
| Volcanic | 6.6 |
| Sea and Ocean | 6.6 |
| Wildfire | 4.1 |
| **Profession** | |
| Scientist | 86.9 |
| Consultant | 5.7 |
| Practitioner | 4.9 |
| Policymaker | 1.6 |
| Scientific communicator | 1.6 |
| Student | 1.6 |
| **Education** | |
| PhD or other postgraduate specialization | 68.9 |
| Master's degree | 27.0 |
| Bachelor's degree | 4.1 |
| **Geographical area of residency** | |
| Europe | 68.0 |
| North America | 11.5 |
| Asia | 5.7 |
| South America | 4.9 |
| Middle East | 1.6 |
| Australia & Oceania | 0.8 |

79          Did not answer          7.4

## 2 The voices collected

The responses to each of the four questions have been categorised into two groups: related to (i) natural hazards (dark grey in Figure 1) and (ii) professional development (light grey in Figure 1). This division is because respondents oriented their answers based on personal judgment, progressed professional experience, and cognitive and emotional background. In the following chapters, direct quotes of responses received are identified with ID and a sequential number (from 1 to 122 for each question). The categories for each question and the related percentage of responses are also included in the Supplementary Material in the form of a Table.

### 2.1 Natural hazards biggest challenges

Natural hazards and disaster reconnaissance have been widely investigated among professional, government, and academic experts. Somewhat lesser is the state of the arts regarding the natural hazards community's grand challenges to direct new approaches for investigation. For this reason, we asked our respondents to express the most critical challenge in natural hazards research (Q1) with no limiting context. The importance of starting from global to local (from natural hazards in general to gender equity and personal experience) aimed at helping the interviewee to get into the topic and value their professional knowledge and expertise about natural hazards. In addition, despite the question being explorative, we wanted to check whether women would have connected the biggest challenges of natural hazards to broad concepts of vulnerability, fragile communities, vulnerable groups, and similar. This is because it has always been one of the greatest stereotypes associated with women (i.e., the most dedicated to caring activities and fragile). Instead, the most perceived challenge (44.3%) is related to climate change and extreme events, focusing on the difficulties of long-term forecasting and predictive models due to the interchange of anthropogenic impacts on the environment.

Similarly, in Frontiers, Wartman et al. (2020) found that computational simulation and forecasting are essential tools for decision making and planning, but they still represent a challenge to the professional community. This result evidences that women professionals in natural hazards do not differ from their counterparts. None of their possible more prominent caring attitudes and sensitivities can affect their perceptions of their work priorities and directions. To continue, respondents believed that one of the most evident constraints is the high complexity and data requirements for model development to provide a reliable forecast concerning the short observation periods, which increases uncertainty. As evidenced by the 10% of the sample, problems with data are multifaceted, and data quality, accessibility, and transparency are an utmost priority. This is especially true when *"research solutions are [...] translated into operational procedures [...] without considering the actual legal framework or the availability of data, referring to a resolution [being too small or too large] that in practice is not used by the managing authorities" ID84*. This mismatch can generate *"[...] confusion among practitioners and managing authorities"* with difficulties harmonising the results and consequent miscommunication risks. Uncertainty is considered a prominent issue in this

regard, especially concerning the unpredictability of climate change as widely acknowledged among scientists. These
are challenging communication efforts, especially when communities lack trust in authorities' decisions or due to
competitive objectives and interests.

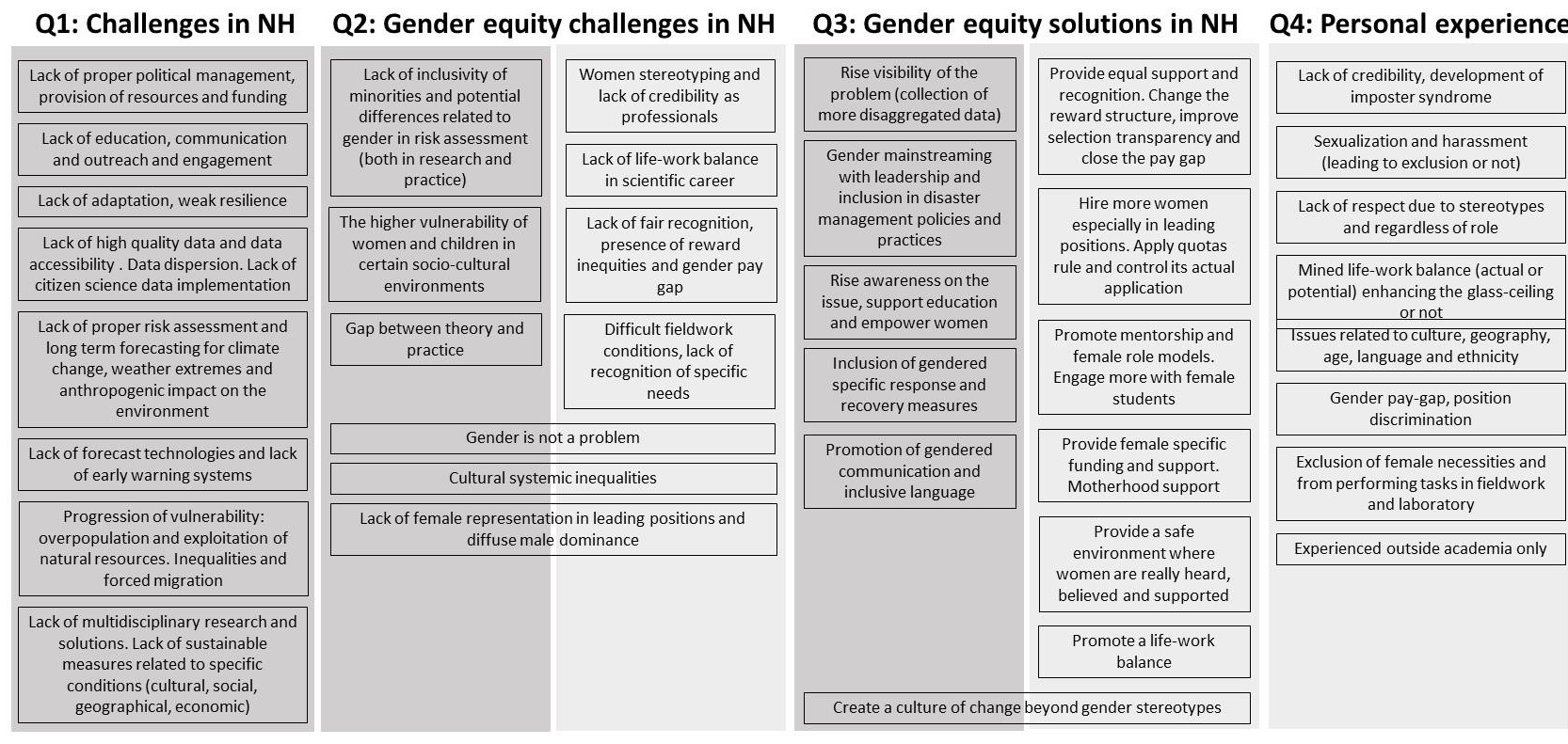

**Q1: Challenges in NH**

- Lack of proper political management, provision of resources and funding
- Lack of education, communication and outreach and engagement
- Lack of adaptation, weak resilience
- Lack of high quality data and data accessibility . Data dispersion. Lack of citizen science data implementation
- Lack of proper risk assessment and long term forecasting for climate change, weather extremes and anthropogenic impact on the environment
- Lack of forecast technologies and lack of early warning systems
- Progression of vulnerability: overpopulation and exploitation of natural resources. Inequalities and forced migration
- Lack of multidisciplinary research and solutions. Lack of sustainable measures related to specific conditions (cultural, social, geographical, economic)

**Q2: Gender equity challenges in NH**

- Lack of inclusivity of minorities and potential differences related to gender in risk assessment (both in research and practice)
- The higher vulnerability of women and children in certain socio-cultural environments
- Gap between theory and practice
- Women stereotyping and lack of credibility as professionals
- Lack of life-work balance in scientific career
- Lack of fair recognition, presence of reward inequities and gender pay gap
- Difficult fieldwork conditions, lack of recognition of specific needs
- Gender is not a problem
- Cultural systemic inequalities
- Lack of female representation in leading positions and diffuse male dominance

**Q3: Gender equity solutions in NH**

- Rise visibility of the problem (collection of more disaggregated data)
- Gender mainstreaming with leadership and inclusion in disaster management policies and practices
- Rise awareness on the issue, support education and empower women
- Inclusion of gendered specific response and recovery measures
- Promotion of gendered communication and inclusive language
- Provide equal support and recognition. Change the reward structure, improve selection transparency and close the pay gap
- Hire more women especially in leading positions. Apply quotas rule and control its actual application
- Promote mentorship and female role models. Engage more with female students
- Provide female specific funding and support. Motherhood support
- Provide a safe environment where women are really heard, believed and supported
- Promote a life-work balance
- Create a culture of change beyond gender stereotypes

**Q4: Personal experience**

- Lack of credibility, development of imposter syndrome
- Sexualization and harassment (leading to exclusion or not)
- Lack of respect due to stereotypes and regardless of role
- Mined life-work balance (actual or potential) enhancing the glass-ceiling or not
- Issues related to culture, geography, age, language and ethnicity
- Gender pay-gap, position discrimination
- Exclusion of female necessities and from performing tasks in fieldwork and laboratory
- Experienced outside academia only


Figure 1: Summary of the categories of challenges and solutions in natural hazards (NH) related to gender equity and personal experiences. In dark grey, natural
hazards related responses, while in light grey, professional and career development related responses.
Enhancing communication is on the top priorities for 17 interviewees (13.9%), highlighting that *"our biggest*
*challenge as scientists is to convince the general public and politicians about our scientific findings and to be able to*
*communicate them properly, in a language that they can understand" ID30*. Problems with comprehension may also
derive from a *"lack of consensus concerning basic definitions (hazard, risk, vulnerability, resilience), leading to*
*misunderstandings or misuse of these terms" ID52* that can affect authorities who can neglect the information received.
27% of interviewees also pointed to a lack of proper political management and insufficient resources and funding. In
this regard, it is even more prominent the need for a *"[...] stronger dialogue between scientists and governments, [for*
*the] identification of strategies and solutions that might be effectively implemented in the real world, thus promoting*
*a research that might really contribute to the solution of real-life problems and not remain in the academic discourses*"
*ID60.*
Integrating multidisciplinary perspectives into this dialogue would significantly enhance the approach
(methodological and communicational) towards such a complex field of research, which 27.9% of respondents
believed. Respondents also indicated a lack of multidisciplinarity, with a concurrent lack of transversal competencies
and integrated solutions for multidimensional problems. Integrating multidisciplinary perspectives into this field
would significantly enhance the approach towards such complex phenomena. Multidisciplinary in natural hazards
means "[...] *build and use land planning integrated multi-risks models which are able to contain both multi-hazard*
*analyses (including hazards evolutions due to climate change) and complex exposure elements (including population*
*migration, natech components)" ID33,* that *"deal with the underlying conditions that influence (social and physical)*
*vulnerability to natural hazards, namely, poverty and inequality" ID37.* This may be well explained by Diekman et
al. (2015) that analysed women's motivation for undertaking a STEM career (for study or work). Collaborative goals,
such as translating theory into practice to help communities advance and enhance development, traditionally appear
to lack in the STEM fields. Inter- and transdisciplinary research may therefore be a women's professional requirement
to be able to consider the multifaceted nature of the problem. However, although it is widely recognised, it is still very
much concentrated within specific disciplinary areas (Latour, 2004). Datta (2018) also recognised the need to
overcome dynamic notions of static disciplinary practice welcoming interdisciplinary research training to solve and
understand the practical challenges from various perspectives. In this regard, we need to *"[...] step outside western*
*norms" ID27*, and the influence that cultural and social relations and power may have on our approach to research:
"*[...] I think that in natural hazards and Earth sciences, in general, we are suffering from a crisis of (lack of) diversity.*
*I think there are many reasons for this. Some are historical, and we can hope that they begin to change as the*
*conversation around diversity becomes more open [than it is now], but some are cultural. Academia does not always*
*foster an environment where these open discussions can be had, and where people are held accountable for their*
*actions" ID98*; thus, a strong connection with collective and policy responsibility exists. Datta (2018) referred to
indigenous knowledge. However, we believe we can expand the discourse to collaborative research knowledge that is
culturally appropriate, respectful, honouring, and careful of the local community promoting anti-racist, gender-
inclusive theory and practice, cross-cultural research methodology, critical perspectives on environmental justice, and
land-based education.
The call for a more inclusive and ethical science that is useful, usable, and used (Aitsi-Anselmi et al., 2018) is
prominent among the respondents and ascribable to the progression of vulnerability investigated and underlined in the
last decade of research in natural hazards and disaster management. Vulnerability but also the progression of
vulnerability for multiple interactive factors is challenging for 16.4% of respondents. A response recognised such
"*[...] underlying conditions that influence the social and physical vulnerability of natural hazards, [are] poverty and*
*inequality*" *ID37*. The representation of women in disaster risk management, who are mostly *"[...] invisible and are*
*not heard*" *ID95*, but also *"women in science and leading positions are still a minority, and therefore their*
*performance and opinions are also sometimes underestimated*" *ID41* (see chapter 2.2 and 2.3). Two respondents
believe that the increased impacts of global warming and the concurrent increase in weather extremes can have an
impact on the most vulnerable individuals globally, *"[...] seeing more [environmental] migration*" *ID79* and *"[...]*
*lead[ing] to [a] reorganisation of populations*" *ID80*. However, despite the financial investments towards natural
hazards mitigation infrastructures, there is much consensus that they are still not evenly distributed, *"even within*
*wealthy nations*" *ID79*. Adaptation, resilience, and sustainable solutions are challenging for the 18% of respondents,
who reported significant obstacles in creating a culture of risk (by increasing awareness) because some natural hazards
cannot be prevented, as they are natural geomorphic processes. Is *"[...] the human behaviour in responding to a*
*natural disaster [that] can make the difference*" *ID86*. Not only, a respondent stated that it is a challenge to *"address*
*inequities for people in [the] location of hazards, access to mitigation/adaptation/preparation/recovery resources,*
*access to hazard warnings, research/observing near underserved communities*" *ID103*; but also *"rather than the*
*technological progress the biggest challenge is reducing the losses where resources are not available*" *ID93*. The last
13.1% argue instead about the poor forecast of hazards, poor understanding of the complexity of phenomena
occurrence and their effects, and lack of early warning systems.
**2.2 Natural hazards and gender equity: challenges and solutions**
Natural hazards affect individuals without fixed distinctions of their gender, and it is important to not over-generalise
a popular trend that sees women vulnerable per default. However, case-specific disaster losses demonstrate how
women and girls are more likely to be disproportionately affected by disasters during and in the aftermath of disasters,
a situation exacerbated by the increase of climate change-induced hazardous events (Neumayer and Plumper, 2007;
Fatouros and Capetola, 2021). The impact includes unprecedented challenges regarding health and well-being, for
example, high rates of mortality and morbidity, prolonged psychological distress, and exposure to high-risk domestic
environments (Fatouros and Capetola, 2021; Thurston et al., 2021)[1], also hampering their opportunity to gainful
employment after the occurrence of a disaster. Socio-economic conditions and cultural beliefs, social norms, and
traditional practices contribute to the complex progression of the vulnerability of women in the wake of natural hazards
and disasters, recognised by 12.3% of respondents. Cultural, systemic inequalities emerge especially in *"[...] lesser-*

[1] Disclaimer: the topic of wellbeing, gender and natural hazards related to psychological and physical burdens (e.g., violence or suicide in the aftermath of a disastrous event) has not been included in the current manuscript because of the lacking competencies to develop such complex clinical topic. In addition, none of the respondents considered this topic in their answers.

*developed countries, but almost everywhere [where] women are paid less and thus have less to respond to disaster*s"
*ID45*. In addition, it is more difficult for a female-headed household to acquire financial assistance and loans that are
essential in the post-disaster rebuilding and re-establishing processes (Alagan and Seela, 2011; Fatouros and Capetola,

189 2021).

Systemic inequalities are also perceived at the family level, because as a respondent expressed, *"women are less*
*encouraged to take information on their own, in most cases, they listen to their partner and agree with their decisions"*
*ID82*, which is not new in literature (Cvetkovic et al., 2018). Patriarchal families can experience communication
problems within the domestic sphere and in the wake of natural hazard occurrences (Cvetkovic et al., 2018; Thurston
et al., 2021). In this context, a respondent added, *"[...] the most obvious challenge is the need to find ways to give*
*women a voice in some countries where, again, the society is male-dominated. Women will often be the people in the*
*household responsible for preparedness and planning activities related to natural hazards. Yet, their opinion may not*
*be sought when decision and policymakers put together plans for improving household resilience" ID109*. Another
respondent, in fact, imperatively stated, *"educat[e] women to react and survive. The experience of the Indian Ocean*
*tsunami 2004 is that women died more than men because they waited at home for their husbands to leave their homes"*
*ID91*. In practical terms, 18.9% of the respondents asked for more awareness and support for educational and
empowerment activities for women. *"Women have unfortunately globally [fewer] opportunities for education and*
*might therefore already be running behind in their understanding of natural hazards and how to prepare themselves*
*and their communities. More effort should be done to reach female communities and educate them" ID104,* expressed
a respondent sharing the concerns of many others who additionally argue for *"[...] enhanc[ing] the connection of*
*women in the field of natural hazards and make their voice heard" ID19*.
The concept of unheard voices is well experienced personally by most respondents and is found in chapter 2.3.
Awareness should not be considered just a means but also a place. We found an interesting comment of a respondent
asking for *"[...] the creation of safe spaces to consider fully the impacts on women in the event of hazard events, and*
*their experiences and frustrations as researchers"ID27*. This approach recognised the need for a horizontal space of
dialogue in DRR, where no top-down or bottom-up approaches are considered. Women's accumulated skills,
experiences, and capabilities in times of natural catastrophes are often not adequately identified, recognised, and
promoted. Women's participation in DRR decision-making processes at all levels throughout the world is meagre. In
this respect, 18% of respondents perceive a lack of inclusivity (of minorities in general, thus extending the vulnerable
pool) and potential differences related to gender in risk assessment (both research and practice). Inclusivity has been
advocated to be *"[...] not just to reach a quota and not only if they first have to be more like the majority (e.g., men-*
*like women, rich coloured people)" ID36*. Respondents share the concern that women and other gender minorities do
not have a seat at the table when it comes to disaster risk management and resilience. Hence, their needs and interests
are excluded from disaster management programmes (Dominey-Howes et al., 2014; Gaillard et al., 2018; Gorman-
Murray et al., 2018), which fail to recognise their diverse economic, political, legal, occupational, familial, ideological,
and cultural backgrounds (Zaidi and Fordham, 2021), creating many issues during response and recovery stages
(Hemachandraa et al., 2017; Thurston et al., 2021). However, women are considered agents of change with unique

skills, qualities, and expertise benefitting quality governance (Gurmai, 2013) through accuracy and transparency in the decision-making process (Araujo and Tejedo-Romero, 2016). Gender inclusion in DRR is recognising and welcoming differences rather than accepting homogeneous thinking. Respondents' testimonies make us realise that the personal experiences in DRR research and management are well integrated into individuals' cognitive and experiential backgrounds. 31% of respondents argue for gender mainstreaming with leadership and inclusion in disaster management policies and practices. They recognise female underrepresentation in leading positions and male dominance in decision-making bodies and communities related to the disaster cycle (18.9%). A respondent is convinced that *"[...] better equity between genders in governing bodies would modify the decision trees of the authorities, particularly in terms of mitigation and long-term view pattern[s]" ID33.*

6.6% of respondents to question Q2 believe that gender is not a (big) problem in natural hazards. Most of their responses refer to positive personal experience in their professional career and the opinion that *"[...] science is likely one of the field[s] that suffers least of gender un-equality. At least in the western countries. [...]" ID86.* Interestingly, none of these eight respondents considered gender an important variable in the disaster assessment or its vulnerability construction. We discuss more about positive changes experienced by the respondents in terms of gender equity in the professional sphere in chapter 2.3.

All the above demonstrates a literature gap in identifying the ways to improve the role of women in disaster risk governance derived by a gender data gap that still exists. 7% of the respondents found it a priority to collect more disaggregated data to raise the visibility of the problem when assessing risks and adaptation options of natural hazards, recognising gender differences without mainstreaming the stereotypes. That might give the idea of gender to be merely connected to a vulnerable condition (Roder et al., 2017) and to be exclusively related to women, promoting stereotypical notions of women as "victims" or the "weaker sex" (Zaidi and Fordham, 2021). This is because, often, vulnerability assessments do not emphasise the fact that individuals simultaneously belong to multiple and intersectional social groups - gender being just one of these - from which they draw their identities and which shape their risk profile in the context of disasters (Zaidi and Fordham, 2021). Real progress towards gender mainstreaming into DRR needs a cultural change beyond gender stereotypes (13% of responses). Possibly, *"[...] it would be great if there could be some overarching guiding principles that all institutions could adhere to, but academia is quite fragmented, so I think it really comes down to individual institutions fostering open conversations and using these to drive change" ID86.* Education is still considered at the base of the change, able *"to build bridges [and] not barriers between each other and to see the richness in diversity and inclusivity" ID112.*

Finally, the need to include gender-specific response and recovery measures is an utmost priority for 4.1% of respondents, where 0.8% argue for a gendered and inclusive language and communication. So, by combining multiple concepts brought up by the interviewees: we need women, and we need to use appropriate language when including them in the DRR policy and practice. However, which women should be involved? This is the interesting question that Enarson (2009) expressed in one of the latest books. She recognised the need to consult and involve local women's organisations and networks, including development and grassroots organisations active in high-risk areas.

We can conclude shortly that there is no 'silver bullet' to solve gender equity in natural hazards. However, there is a
need to know how useful and effective concrete examples, specific suggestions, action guides, and indicators are to
mainstream gender into DRR.

**2.3 Professional development and gender equity**

The questions related to natural hazards and gender equity (Q2 and Q3) had been received to be related to natural
hazards per se (see chapter 2.2) and for some others to professional development (Figure 1, light grey boxes). Only
Q4 specifically addressed gender-based issues in the work environment; in particular, we asked for personal
experiences. Since personal experiences and general challenges often coincide, we have used both to address the
abundant issues still residing within the community and the actions to be implemented for a more inclusive work
environment. The challenges perceived in natural hazards and gender equity (Q2) are for the 37.7% of responses
related to the lack of role models and female representation in decision roles and leadership positions, showing the
range of career possibilities and paths. In addition, 36.1% of respondents (Q2) evidenced unresolved challenges related
to an unfair reward structure, pay gap, life-work imbalance, stereotyping and lack of recognition in a male-dominated
field. However, these are not just perceptions, but they are matched by 73.8% of personal experiences (Q4), who have
confronted career advancement and unfair treatment obstacles.
In detail, 27.9% experienced being attributed a lower salary compared to male colleagues and being discriminated
against obtaining leadership positions: "*[...] More visibility is given to male colleagues all the time. Even more power*
*and resources are given to them. In my place of work (State organisation), power positions belong 100% to men, [...]*"
ID17. Moreover, 14.8% of respondents also experienced or witnessed life-work imbalance particularly worsened due
to unequal expectations of women and men's family responsibilities. A respondent reported that "*it has always been*
*very difficult to combine motherhood with the challenges of making a career [...]*" ID37 and another echoed that "*it*
*has been very hard to find role models in my field when I took the decision of having a family. I had no reference for*
*a successful woman in my field with children [...]*" ID69.
Unfair treatment has also been experienced widely by our respondents. A respondent reported, "*My opinions have*
*been quite often undervalued by other colleagues. Even when I was the PI of a project, some people preferred to speak*
*to male colleagues*" ID110. Compared to male colleagues, a lack of credibility was reported by 27.9%, a lack of
respect regardless of role by 23.8%. Sexualisation and harassment were reported by 13.9%. One of the interviewees,
unfortunately, shared one of the most negative experiences: "*[...] Anything deemed "feminine" about me was used*
*against me as a weakness. Constant inappropriate talk [was] designed to see if it would get a reaction out of me by*
*my co[-]workers. In the field, free time was spent at the bar or even hostess lounges, and I was incredibly*
*uncomfortable [...]. Then I was put in a closed-door meeting with just my supervisor and asked how working there as*
*a woman was. I felt very unsafe and therefore unable to be truthful [...]*" ID79. Discrimination can be so pervasive to
induce repression of one's traits, to the point of feeling *"[...] pushed to be more "masculine" in the workplace to fit*
*in"* ID79. To our dismay, the biases and stereotypes reported, and the harassment experienced are not new to women
working in male-dominated disciplines or literature (Kenney et al., 2012), news outlets and documentaries (Picture a
Scientists, 2020). Despite the wide recognition of the problem, progress is still slow. Cultural, systemic inequities are
part of this problem and are linked not only to gender stereotypes but also to age, ethnicity, religion and nationality
(9.8% of respondents).
Finally, 8.2% of respondents reported issues related to fieldwork: they experienced exclusion and lack of consideration
of their specific needs precluding them from performing tasks. In some cases, the problem is again very much related
to performing capabilities stereotypes; one respondent reported, *"[...] Many times in the field I was asked, "are you*
*sure you can do this (going uphill, going down, dirt myself)? [...]" ID44.* But also feeling uneasy *"[...]  about certain*
*accommodations (e.g., bathroom) that I feel I might be imposing on my peers, and thinking twice about taking valuable*
*measurements in areas where my safety might be at risk" ID101.*
A positive trend has been observed concerning structural changes in recent times. For example, one respondent who
experienced discrimination in the past recognised that *"[...] female colleagues entering the field now, with solid*
*competencies and a lot of "guts", have much more chances now to move up to decision positions [...]" ID23.* In
addition, 23% of respondents explicitly said they did not experience any gender-related career challenges reporting
their positive experience in a supportive environment and gender-mixed teams (both at the educational and the
professional level). Although for a couple of respondents, the personal experience was positive, they reported being
aware of gender-related challenges encountered by other female colleagues.
We can conclude that the struggle for women to find inclusive work environments was and still is not resolved, despite
recognising positive efforts in the right direction and some virtuous examples. Solutions concerned with promoting
gender equity in the work environment are envisioned by 54.1% of the responses to Q3. The proposed solutions will
not read unfamiliar to those accustomed to the debate in the broader gender-related STEM career challenges:
*"Diversity begins at the top. Work to understand why retention is challenging and change reward structures. Put*
*women in leadership positions. Refuse to hold all-male panels, all-male sessions, all-male anything" ID42,* said one
respondent, well summarising the general feeling of the interviewees.
43.9% of responses suggested enhancing selection transparency via providing equal support and access to resources
and information, recognising women's work, and changing the reward structure, ensuring an experience-based salary
to close the gender gap. Bell and co-authors advocated for such changes and actions almost 20 years ago (Bell et al.,
2003). It is noteworthy and disappointing how slow the process to equity is if we still discuss the benefit these changes
would accomplish today. Indeed, many institutions have taken steps forward in these regards. However, the mission
is far from being complete, and possibly one reason is that the efficacy of actions undertaken is often not measured or
not publicly shared (Timmers et al., 2010; McKinnon, 2020). Promoting women's work reflected 31.8% of responses
calling for hiring more women, particularly in high profiles and relevant positions, as a solution. To achieve that,
quotas are one of the actions commonly proposed. Quotas have been since long introduced in many institutes and
funding organisations and resulted in an effective reduction of the gender gap in leading roles in certain areas (Handley
et al., 2020; Pellegrino et al., 2020). However, as also some respondents noted, quotas rules may appear only on paper
at times. They may also be seen as controversial or counterproductive, reinforcing old stereotypes (Handley et al.,
2020, Pellegrino et al., 2020). We believe that quotas can be a double-edged sword able to raise negative opinions
among women in the workplace, undermining their credibility. However, quotas can be a valuable instrument to
promote and normalise more gender balance environments until more transparency in selection procedures is enacted.
One respondent, for example, pointed out, *"[...] as a woman, I am always extremely disappointed when positions are*
*open only for my gender. First, because it means that male[s] in this specific institution had the power to only employ*
*other males. Second, because women employed at such positions can always be taught that they only got it because*
*of their gender, not their capacities" ID12*. A global survey targeting Earth and Space scientists by Popp et al. (2019)
clearly showed the divided opinion on quotas. They noted how quotas' favour tends to be gendered, with 44.9% of
women and 27.9% of men sharing a favourable opinion and career stage related. Among women favouring quotas,
56.1% are postdocs, while among men the 34% hold a professor position. They concluded this result showed a clear
sign of a disadvantage for early-mid career women and a fear of being negatively affected by quotas for mid-career
men geoscientists (Popp et al., 2019). Handley et al. (2020) have analysed the gender balance in universities in
Australasia and noted that even if quotas regulations were in place, few-to-no women would apply to vacancies for
various reasons. Therefore, to counteract the issue, they proposed creating a database of female professionals working
in geosciences divided by area of research. Such a database can be used to find new collaborators, advertise vacancies,
and invite applications from relevant candidates (possibly leading to a larger number of female applicants), inquire
about consultancy, ask for an interview, and pool for surveys. We find this solution interesting and responding to the
needs of giving equal career opportunities while maintaining a transparent process and recognising female
professionals. Such a database could also be used to promote female-specific mentorship and role models, including
increasing the visibility of women's work and thus help engage more female students and potentially retain them in
the field, as noted by 27.8% of responses. On mentoring and role models, Handley et al. (2020) highlighted an
important point. Since not many women occupy apical positions yet, horizontal mentoring among women peers or
close in the career stage can also be a good option. For several years, several associations have made their primary
goal providing support and mentoring to women in geosciences. To cite a few at the international level, the 500 women
scientists established in 2016, the Earth Science Women's Network (ESWN, Adams et al., 2016) and Geolatinas
founded in 2002. A complete list of women-focused and women-led geoscience and related networks are available in
Handley et al. (2020). Moreover, female-specific funding and support schemes, including those specific for supporting
motherhood, are solutions for 21.2% of respondents. The latter goes together with the promotion of life-work balance,
the acceptance of part-time careers and a better redistribution of roles and responsibilities, which are seen as significant
help by 13.6% of responses. In addition to promoting more women in our work environments and provide adequate
support, institutions must become safe places where people in "[...] *positions of power and administration take*
*harassment claims seriously and stand by a zero-tolerance policy and made women feel comfortable and believed*
*when reporting these issues" ID80*, said a respondent, reflecting the 15.2% of responses.
We can conclude that one of the main steps forward with the potential of a profound impact resides in a broad cultural
change that will break down those still longing stereotypes and allow real diversity inclusion. 27.8% of responses
explicitly hope for this change in the work environment, but it is possible to include all actions proposed in this much
broader resolution. Cultural changes are slow to achieve. Keeping up a constructive debate and the attention around
the topic helps as much as the proposed change in the reward structure, the promotion of women's work, hiring more
competent women for apical positions, providing motherhood-specific support and redefining roles and
responsibilities. We do not exclude the immense necessity towards the normalisation of co-parenting and genderless
or gender equivalent parental initiatives. We believe that there are very prominent actions undertaken in this direction
in some countries. However, they are political regulations where we, singularly, have little to no control. Instead,
institutions (or companies) can lead the change and become the first promoters of equal support with well-thought
plans and effectiveness assessment.
One more way to foster profound changes passes by promoting inclusive language at all levels, particularly from
people in leadership positions, regardless of their gender. Language shapes profoundly our mind, our way of
interpreting the world we live in, the words we use can discriminate as much as they can empower (McKay et al.,
2015; Taheri, 2020). Where not yet in place, specific training on inclusive language and unconscious bias should be
organised at institutions and organisations and possibly be made mandatory with a top-down priority.
The solutions envisioned by the pool of respondents to our survey are very similar to strategies already highlighted in
the literature, reported in Table 2. We can conclude that strategies, actions, and solutions are well defined and, in some
instances, already enacted. However, monitoring the efficacy of these actions is far more complex but of great
relevance to understanding which of them is worth pursuing and which instead do not provide significant improvement
towards closing gender-based issues. Timmers et al. (2010), analysing aggregated data for employment in the year
2000-2007 in 14 universities in the Netherlands, could observe that the larger the number of gender equality policy
actions adopted, the more significant the reduction of the glass ceiling. However, they criticised the lack of internal
evaluation of the adopted measures by the universities themselves. Universities, research institutes and organisations
should promote researching and applying adequate methods for monitoring their strategies and implementing them
with high priority.
*Table 2. Summary of strategies and envisioned solutions towards gender equity in STEM and geoscience from recent*
*literature and this study. It can be observed how the proposed solutions align well among themselves showing strong*
*similarity, when a solution has been proposed that does not find direct comparison the related box is left blank.*
*\*Handley et al. (2020) focus mainly on the Australasia situation. However, these data are fundamental to be also*
*gained elsewhere in the world.*

| Vila-Concejo et al. (2018) | Popp et al. (2019) | Handley et al. (2020) | This perspective |
|---|---|---|---|
| Redefine success | Transparent candidate selection criteria of institutions and funders for hiring processes and funding opportunities | Re-think excellence recognition and reward criteria | Provide equal support and recognition. Change the reward structure, improve selection transparency, and close the pay gap |

| | | | |
|---|---|---|---|
| Advocate for more women in prestigious roles | Better promotion and representation of female scientists by selecting them for prestigious decision-making roles in scientific organisations and institutions | Raise the visibility of women through open-access databases | Hire more women especially in leading positions. Apply quotas rule and control its actual application |
| Encourage more women to enter the discipline at a young age | | Greater promotion of the value of mentoring and provision of inclusive mentoring programs | Promote mentorship and female role models. Engage more with female students |
| Create awareness of gender bias | Mandatory gender bias training to combat unconscious biases | Engage all the geoscience community to create sustainable change | Create a culture of change beyond gender stereotypes |
| Get better support for the return to work | Granting more rights, flexibility, and support for parents to share parental responsibilities and to transform academia into a more family-friendly workplace | | Promote a life-work balance |
| Promote high-achieving female | | | Provide female specific funding and support. Motherhood support |
| Speak up | | Eliminate and actively address everyday sexism and harassment in geosciences: Field trip code of conducts | Provide a safe environment where women are really heard, believed, and supported |
| | | Gather more data on why women leave geosciences* | |
| | Inviting more men to an open discussion about gender equality | | |


## 3 Getting down to business

From the responses analysis and state of the art literature, we have understood that gender-based challenges at the
professional level and within the disaster cycle are very close. Moreover, because of their interrelation, the solutions
proposed may not be exclusive for a professional or a more technical sphere, but they can work simultaneously, with
mutual benefit. Early education is key to fostering a cultural revolution. If children attend classes related to social
norms, diversity, and inclusion, they might become adults able to go beyond individuals' gender. If so, women and
other gender minorities would be much more considered at the leading positions in DRR institutions or academia, thus
promoting a more comprehensive vision about vulnerabilities before, during, and after natural hazards occurrence.
But the cultural change must also be vertical in a top-down approach by organising specific compulsory training for
leaders and professionals to explain biases and stereotypes and fight them to promote a more effective and just natural
hazards management and, thus, a more inclusive society. In addition, the scale of the change should consider the
horizontal space in which role models are found within peer networks to promote and support positive imitative
behaviour.
For what concerns the guiding principles and institutions, several examples highlighted in this perspective showed
how the political agenda (e.g., SFDRR) lacks any gender-related practical guidance. So do all other local
administrations and institutions. Many gender-inclusive initiatives are short-term and aim primarily to spark interest
rather than build skills. Most of the time, they are just a box 'ticked' rather than an effective action. Therefore, we
advocate for compulsory study, implementation, and application of methods to measure and monitor over time the
efficacy of actions and strategies put in place at institutional, national and international levels.
In addition, current gender-inclusive initiatives are excluding men despite literature demonstrating a disjunction
between the assumptions and lack of understanding of the reality of men's lived disaster experiences (e.g., Rushton et
al., 2020). What Fordham and Meyreles (2014) called a paradox, masculinity, which contributes to the structure of
power that privileges men, can also put men at risk (e.g. Jonkman and Kelman, 2005; Ashley and Ashley, 2008;
Fitzgerald et al., 2010). Similarly, we can observe how in the professional domain, specific jobs and disciplines are
still perceived as belonging to a (stereotyped) female world only and where men are seen as outliers. If the final goal
is a truly inclusive society, we must be aware of all the biases and stereotypes we are surrounded by and counteract
all of them appropriately. The future of research in natural hazards and disaster mitigation and our professional domain
needs to include all voices and find allies in the privileged categories of the specific domain of interest. We think that
lessons learnt within the context of women discrimination can serve as starting point to expand the discourse to other
gender minorities and that intersectional research should be advocated for to gain an all-inclusive approach and
understanding of disaster stories that foreground differences.
**5. Authors' contributions**
All authors have contributed to the Conceptualization and Data curation. VC and GR have equally contributed to the
analysis and preparation of the first draft. All authors have contributed to the revision and editing of the manuscript.
**6. Competing interests**
Author HK is executive editor of the journal NHESS.
**7. Special issue statement**
The manuscript is submitted as part of the Special Issue "Perspectives on challenges and step changes for addressing
natural hazards."

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
