# Peer review of "Invited perspective: “Natural hazard management, professional"

_Natural Hazards and Earth System Sciences, 2021_

## Author Response (AR1)

We wish to express our appreciation for both Reviewers' in-depth comments, suggestions, and corrections, which have greatly improved the manuscript. Each point raised has been answered, and the answers are highlighted in bold. Reference to lines refer to the marked version of the revised manuscript.

**Reviewer 1**

In this perspective piece, the authors present and discuss the challenges related to gender within the natural hazards field on the bases of survey data. First of all, I want to congratulate the authors on this extremely relevant and well written piece. The article touches upon many issues around gender always in a critical way, which is what I appreciate the most about the piece. In my opinion, this article should be accepted for publication in NHESS almost as is, but there is one clarification needed (which could play a rather large role for the meaning of this paper):

Was the survey aimed at women and men or only women? If both but only women replied, what does this tell us? If this is the case, I believe this is an extremely important point that should be discussed in the paper: e.g. why men did not take part in the survey? What could be the reasons? If it was only aimed at women, what were the reason? Could there be a risk of losing data on how men perceive (or don't perceive) this issue? I think then, in such case, this should be discussed. I am asking this because in p.2, line 46 it says "opinions of individuals working in the broad field of natural hazards".

**Regarding your first question, the survey aimed at people identifying with genders, which are a minority in the field of natural hazards, i.e., mainly women, but also responses by non-binary individuals were considered. The reason is that we wished to highlight the usually unheard voices. The first "barrier" question was about which gender the responder identified with. If the answer was male, the survey stopped for the respondent; if the answer was female or non-binary, the respondent could continue to answer. As such we collected 121 answers from female respondents and 1 answer from a non-binary respondent.**

**We realised we hadn't addressed this point very clearly; therefore, we rephrased lines43-48 as follows: "In this 'invited perspective', we have put individuals identifying themselves with genders that are a minority in the field of natural hazards, i.e., female and non-binary genders, at the centre of the discussion. We aim to concretely contribute to understanding the standpoint of these minorities who are often underrepresented, unheard and poorly considered professionally and in DDR policy and practice. Thus, this perspective qualitatively explores a collection of 121 opinions of individuals identifying themselves as female and one opinion of an individual identifying themselves as non-binary working in the broad field of natural hazards (in academia, in the industry, as practitioners or policymakers)."**

In relation to the former point, how was the survey advertised? I think a few lines about this would help clarify the first point

**The survey was distributed via email through Natural Hazards related mailing lists, NHESS Authors via Copernicus and a list of female professionals collected by the Authors through their networks. Moreover, the survey was advertised on social media, particularly Twitter, LinkedIn, and Facebook from the personal accounts of the Authors when available.**

**To clarify the promotion of the survey, we added the following text at lines 80-83: "The survey, i.e., link to the questionnaire with a short explanatory and motivational text, was advertised via email to the EGU NHESS author list and to a list of female professionals that the authors had collected in their networks. Moreover, the survey was advertised on social media, particularly on Twitter, through the personal accounts of the first two authors."**

Minor points:

It may be that I couldn't find it, but it seems the abstract is missing (could only find the short summary)

**We have not included an abstract in light of the structure of other NHESS invited perspectives. If the Reviewer or Editor thinks it is useful, we will add a few lines regarding the main objective, method, and general conclusions at the beginning of the paper as abstract. We hope for guidance by the editor, please let us know if we should provide an abstract during the revision of the paper.**

1 line 19: I think it should be "… gender-based issues in DRR (even beyond …)"

**We accepted the suggestion of the Reviewer.**

2 lines 30-31: this sentence needs referencing

**The statement is the result of our personal search over the most common academic web engines. This is not the central core of the discussion, still we wanted to start the presentation of this topic with this broad statement. Thus, a new version of the paragraph has been proposed as follows (lines 33-38): "Based on our literature search, we recognise that for most disaster-related papers, gender was merely used as a dichotomous variable (usually together with a set of other socio-demographic variables) to test assessments and model results, which are the core of the papers."**

2 line 33: I think this was actually discussed broadly also by Finucane et al. (2010), and more recently by Mondino et al. (2021)

**We thank the Reviewer for the suggested references; we added them to the new version of this paragraph.**

2 line 38: "non-technical" articles: do you mean articles in the social sciences sphere? Or articles that use qualitative as opposed to quantitative approaches?

**Yes, we meant articles prominently in the social sciences that are often considered less technical. We clarified this in the new version of this paragraph.**

2 line 56: I think this is a great approach to this type of data analysis!

**We thank the Reviewer for the positive comment.**

2 line 57: related to the first "major" point: was gender asked in the survey?

**Yes, as pointed out above, gender was asked as a barrier question at the beginning of the survey. We also added a new sentence (lines 69-70) stating: "Individuals recognising themselves as male were excluded from the survey via a first barrier question about their gender."**

4 line 84: "Samwise" do you mean "similarly"?

**Yes, we have decided to rephrase with "similarly" for clarity.**

7 lines 167-168: I think this sentence needs a reference

**We added Cvetkovic et al. (2018) also at the end of this sentence.**

7 line 200: "cognitive and experiential […?]" I think a word is missing

**We added the word "background" at the end of the sentence.**

8 line 222: "aroused by" should be "brought up by"

**We have rephrased it as suggested.**

9 line 238: "unbalance" should be "imbalance"

**We have rephrased it as suggested.**

References:

Finucane, M. L., Slovic, P., Mertz, C. K., Flynn, J., & Satterfield, T. A. (2010). Gender, race, and perceived risk: The 'white male' effect. Health, Risk & Society, 2(2), 159–172, https://doi.org/10.1080/713670162

Mondino, E., Scolobig, A., Borga, M., and Di Baldassarre, G. (2021). Longitudinal survey data for diversifying temporal dynamics in flood risk modelling, Nat. Hazards Earth Syst. Sci., 21, 2811–2828, https://doi.org/10.5194/nhess-21-2811-2021

Personal opinion(s)

I would like to use this space to voice some of my personal opinions on the matter. These are not meant to influence the perspective piece in any way, but just to further discuss this relevant and (unfortunately) timely issue. While in general I agree with most strategies proposed in this perspective and in the other articles mentioned in Table 2, I find it hard to agree with the fact that "providing female-specific funding and support" is a good way forward. I am afraid this can very quickly turn into a double-edged sword, as in "she got that funding because it was women-only funding". Such approach in my opinion would (maybe partly) address the symptoms, but without actually curing the disease. What we need in terms of funding, in my opinion, are evaluation panels that are diverse, that are held accountable, and that are trained. I have a similar opinion around quotas, which the authors also mention in their piece. Quotas are indeed controversial, and I found myself discussing their pros and cons many times with female and male colleagues. My personal opinion is that quotas can be detrimental to a woman credibility if she's put there because of quotas. Obviously, this would not be the case all the time, but it's definitely something that can be used against her at any point. I don't think it is worth the risk. At the same time, during discussions with colleagues, another point was brought up, i.e. that the "forced" presence of women through quotas can help normalise and promote gender balance within decision bodies. To be fair, I believe this entails a lot of wishful thinking and optimism.

**We thank the Reviewer for sharing their valuable opinions. We fully understand the point raised, and we agree that quotas can be seen as controversial and that a general, more transparent reward structure could potentially be the way out of quotas. However, until more transparency is enacted, quotas can be a valuable tool as female-specific support.**

**As this comment brought very helpful comments, we decided to include this into the main body, now line 348-353.**

In terms of motherhood support, I couldn't exactly understand what is meant by it in the paper, but again I believe that it's not only about "supporting motherhood" but rather "normalising fatherhood", even better so "parenthood". In my opinion, the intrinsic issue here is that, in most countries, parenthood seems to be a female-only prerogative. Now this would be one of the rare cases in which I would support a sort of "forcing", as in forcing both parties (i.e. both parents) to take an equal amount of parental leave, for example. I say I support this because I've seen it working in countries such as Sweden, where parental leave (maternal and paternal) is absolutely normalised.

**We can only agree with the Reviewer; we also hope to see more countries taking political steps towards normalising co-parenting and genderless or gender equivalent parental initiatives. In the meanwhile, particularly in countries whereby work and parenting regulations for family caring tasks are taken over most if not only by the female part of the couple, we think it is essential that institutions and funding bodies provide female-specific support when needed. We have added this into the discussion, now lines 425-430.**

Anyway, these were just two points that I felt like would be an interesting discussion-starter, and I hope the community will start discussing this seriously. Eventually, it is not about "making it easier" for women, it is about "not making it harder".

**We couldn't agree more with the Reviewer, and we join you in the desire to see more discussion of the topic proposed here within the community.**

**Reviewer 2**

It was great to review this sobering and much-needed article. I really enjoyed reading and think the discussion is well written. I have 3 major comments that need to be addressed before publication.

It is not clear who the target group of the survey was. How was the sampling strategy? Newsletter? Existing networks? Twitter? How many people were invited? It is not clear why only women replied. Did the survey target only women? If this is the case, what are the reasons for excluding men from the survey? Please add this information to the methodology session.

**Yes, the survey target was women and gender minorities; thus, we chose a first barrier question about gender. Only individuals recognising themselves as female or non-binary were invited to continue to answer the questionnaire. We have decided to proceed in this way to raise the voice of those individuals representing a minority in the field of Natural Hazards.**

**We realised we hadn't addressed this point very clearly; therefore, we rephrased lines 43-48 as follows: "In this 'invited perspective', we have put individuals identifying themselves with genders that are a minority in the field of natural hazards, i.e. female and non-binary genders, at the centre of the discussion. We aim to concretely contribute to understanding the standpoint of these minorities who are often underrepresented, unheard and poorly considered professionally and in DDR policy and practice. Thus, this perspective qualitatively explores a collection of 121 opinions of individuals identifying themselves as female and one opinion of an individual identifying themselves as non-binary**

**working in the broad field of natural hazards (in academia, in the industry, as practitioners or policymakers)."**

**To clarify the promotion of the survey, we added the following text at lines 75-78: "The survey, i.e. link to the questionnaire with a short explanatory and motivational text, was advertised via email to the EGU NHESS author list and to a list of female professionals that the authors had collected in their networks. Moreover, the survey was advertised on social media, particularly on Twitter, LinkedIn and Facebook through the personal accounts of the first two authors."**

**In terms of number of people invited, 112 female colleagues from the personal list compiled by the authors, 3085 authors from the NHESS Copernicus database, the gender of the authors is not known. The survey was also advertised on different social media platforms from which is hard to have an accurate estimation of people engaged. We can speculate we reached in total around 5000 people.**

I am not entirely convinced about having the category "main challenges on NH research" discussion on pages 4 to 5. It is out of the scope of the paper. I think it would be better if you linked the identified challenges to gender and diversity issues. It is currently not in the main objectives. Yet, it gets much focus, mainly because it is the first topic you write about. I would consider adding this as one of the paper's goals or toning it down and linking it more with gender/diversity (like it is done on page 6). Especially the part regarding forecasting problems, data quality, etc. are disconnected from the paper objectives. I understand how relevant and exciting these results are and that you want to show them, but as a reader I felt it belonged to another paper.

**We understand the point raised by the Reviewer. We decided to ask that question and thus include this chapter for two main reasons: 1) first, because we wanted to value the professional opinions of women (and other gender minorities) before getting into the topic of gender equity to 2) understand if the global challenges of natural hazards perceived where coinciding or colliding with the most common women stereotypes (e.g., thinking that for example, the most critical challenge would have been related to vulnerabilities worldwide – as individuals recognised as females are mostly considered the most caring and sensitive, for example). We have better contextualised this into the chapter.**

I suggest adding figures with the most common challenges and solutions mentioned by the participants. Also, since you have lots of text data, you could consider making a graph with the most cited terms or even a word cloud.

**We thank the Reviewer for the suggestions, we have tried to make a word cloud, but it didn't result in an obvious picture. Mainly because the open nature of the questions let the respondents express concepts that we could manually categorise recognising synonym or similar expressions. The automatic word cloud algorithm tested couldn't group these similarities appropriately.**

**Figure 1 already summarises the challenges and solutions mentioned by the respondents. Specific percentages are included in the main body of the manuscript and now in a Table in the supplementary material.**

Specific comments

Line 17: It would be better to explain here how Zaidi and Fordham (2021) achieved these findings. Sure, one can always look at the original source, but if you put the information here it is easier for the reader. It could be something like this: "By examining xxx and xxx data, Zaidi and Fordham (2021) found out that the SFDRR….)

**Zaidi and Fordham (2021) reviewed the Sendai framework documents and data outputs to examine the effectiveness of proposed SFDRR gender-based strategies. We rephrased the sentence to take into account the Reviewer's comment.**

Line 29: what is meant here by "(referred to the journal)"?

**We mean that gender and disaster publications are scarce and found in specific journals focusing on the sphere of the social sciences. We have rephrased the sentence as: "Based on our literature search, we recognise that for most disaster-related papers, gender was merely used as a dichotomous variable (usually together with a set of other socio-demographic variables) to test assessments and model results, which are the core of the papers."**

Line 34: There are studies that show that actually men are more affected because of their risk-taking behavior when facing disasters. This is true in some European countries. However, I agree that this kind of binary logic doesn't help much DRR research.

**Yes, we agree with the Reviewer comment. We have touched on this issue also in the conclusive chapter (lines 438-440).**

Line 41: Could you enumerate some of these challenges? The sentence is a bit vague

**The challenges are related to the gender gap index construction variables, including political empowerment, economic participation and opportunity, educational attainment, health, and survival. We included some example into the sentence.**

Line 48: You can remove EU Survey from here, as it is explained in detail later.

**We have removed the sentence and included the information on when the survey was conducted in line 62.**

Figure 1 is too small to read. What were the criteria for including categories here? For instance, only things mentioned by 5 people were included? What guided this analysis? It would be interesting to know how many people support each category and which ones were less mentioned.

**Thanks for the comment; we understand we might have probably not been very clear on this point. We categorised and discussed all opinions collected; thus, the figure shows all resulting categories. We added this information "All categories are shown in Figure 1." on line 92-93. In the text, the representative categories (with their percentage of answers received) are all visible. We included an overview table with all categories and the number of answers they received into the appendix. Regarding its readability, the figure could be made bigger if flipped and occupying a whole page.**

Fig 1: how many people answered that gender is not a problem? It would be good to have a paragraph with the arguments used by these participants. I would be interested in seeing their demographics. Have

you considered applying statistical analysis to your outcomes to see if there are group differences? Or the sample is too small for this?

**The percentage of responses falling in this category is provided in the manuscript's text (line 247 and following), which now states "6.6% of respondents to question Q2 believe that gender is not a (big) problem in natural hazards. Most of their responses refer to positive personal experience in their professional experience and the opinion that *"[…] science is likely one of the field[s] that suffers least of gender un-equality. At least in the western countries. […]" ID86*. Interestingly, none of these eight respondents considered gender an important variable in the disaster assessment or its vulnerability construction. We discuss more about positive changes experienced in terms of gender equity in the professional sphere in chapter 2.3". In no one case, we have given value to the answers just because of higher response rates.**

**We have undertaken statistical analyses to identify group differences, however, no one resulted significant. The open and exploratory nature of the survey resulted in heterogeneous yet disproportionate representation of some demographic characteristics, and in high numbers of categories. Thus, a dataset that is not suitable for statistical analyses (which was also not our intention, since we aimed at a qualitative, explorative analyses). For the category "gender is not a problem", the respondents, in terms of demographics (natural hazard field, profession, education, geographic origin) did not differ from the average characteristics of the whole group.**

Line 84: Samwise?

**We have rephrased with "Similarly".**

Line 155-156: Again, please bring here articles that show the opposite: men are more affected in some cases. Here some possible examples, but there are many more.

Wachinger, O. Renn, C. Begg, C. Kuhlicke, The risk perception paradox-implications for governance and communication of natural hazards, Risk Anal. 33 (2013) 1049–1065. doi:10.1111/j.1539-6924.2012.01942.x.

S.. Cutter, J. Tiefenbacher, W.D. Solecki, En-gendered fears: femininity and technological risk perception, in: S.L. Cutter (Ed.), Hazards Vulnerability Environ. Justice, Earthscam, New York, 2006: pp. 177–192.

S.T. Ashley, W.S. Ashley, Flood fatalities in the United States, J. Appl. Meteorol. Climatol. 47 (2008) 805–818. doi:10.1175/2007JAMC1611.1.

Fitzgerald, W. Du, A. Jamal, M. Clark, X.Y. Hou, Flood fatalities in contemporary Australia (1997-2008): Disaster medicine, EMA - Emerg. Med. Australas. 22 (2010) 180– 186. doi:10.1111/j.1742-6723.2010.01284.x.

S.N. Jonkman, I. Kelman, An analysis of the causes and circumstances of flood disaster 27 deaths. Disasters. 29 (2005) 75–97. doi:10.1111/j.0361-3666.2005.00275.x

**Since we touch this topic in the conclusion chapter, we have added the following references in lines 439-440 Jonkman and Kelman, 2005; Ashley and Ashley, 2008; Fitzgerald et al., 2010.**

These articles could also be helpful for discussing your results:

https://www.sciencedirect.com/science/article/abs/pii/S2212420921004568

https://gh.bmj.com/content/6/4/e004377.abstract

**Thank you for the suggestions. We added these references to our discussion to strengthen some of the points raised. With the last two links suggested, the Reviewer has pointed out a critical issue. We have not included a lot of other gendered behaviours during and after natural hazards occurrences, such as domestic violence, suicide, etc. For this reason, we have claimed at the end of the manuscript that our limiting approach and analysis does not reflect the multitude of relationships (most of the time negative) among gender and disasters.**

Line 157: It would be good if you could back up these statements with concrete examples and empirical research results. Otherwise, it is too vague.

**We have added a few examples and related references to back up this statement. "The impact includes unprecedented challenges regarding health and well-being, for example, high rates of mortality and morbidity, prolonged psychological distress, and exposure to high-risk environments (Fatouros and Capetola, 2021; Thurston et al., 2021), also hampering their opportunity to gainful employment after the occurrence of a disaster."**

Line 169-172: This is a very good point

**We thank the Reviewer for the positive comment.**

Line 220: Did the participants have the option to choose different solutions? Or it was an open-ended question? Please make this clear in the methods section. It would be good to have a graph with the most cited measures/solutions

**The four questions were all open. It is clearly stated in line 56, "We have chosen open questions to let the professionals personally provide…". We have added a table in the supplementary material that includes the categories and related percentages.**

Line 226: Are these results consistent with previous literature? Are there any similarities? If there is no research on gender in NH research, you can perhaps compare with other fields (e.g. mathematics) and see if the problems/solutions are somehow similar or if there is something unique for NH research

**Similarities and consistency with previous literature are compared trough out the chapter for each challenge/solution proposed. For example, in line 232 and following we state "Respondents share the concern that women and other gender minorities do not have a seat at the table when it comes to disaster risk management and resilience. Hence, their needs and interests are excluded from disaster management programmes (Dominey-Howes et al., 2014; Gaillard et al., 2018; Gorman-Murray et al., 2018), […]".**

**Another example is Table 2 comparing the solutions proposed by the respondents to solutions collected by other three studies Vila-Concejo et al. (2018), Popp et al. (2019) and Handley et al. (2020).**

Line 235: again, it would be helpful to have a graph with the most common challenges cited by participants.

**Categories are provided in Figure 1 and also in the newly made Table provided in the supplementary materials where also percentages are given for the different categories.**